# Telomere Length: A Cardiovascular Biomarker and a Novel Therapeutic Target

**DOI:** 10.3390/ijms232416010

**Published:** 2022-12-16

**Authors:** Marios Sagris, Panagiotis Theofilis, Alexios S. Antonopoulos, Konstantinos Tsioufis, Dimitris Tousoulis

**Affiliations:** Cardiology Clinic, ‘Hippokration’ General Hospital, School of Medicine, National and Kapodistrian University of Athens, 157 72 Athens, Greece

**Keywords:** telomere length, LTL, telomerase, cardiovascular disease, myocardial infarction, atherosclerosis, senolytics, treatment

## Abstract

Coronary artery disease (CAD) is a multifactorial disease with a high prevalence, particularly in developing countries. Currently, the investigation of telomeres as a potential tool for the early detection of the atherosclerotic disease seems to be a promising method. Telomeres are repetitive DNA sequences located at the extremities of chromosomes that maintain genetic stability. Telomere length (TL) has been associated with several human disorders and diseases while its attrition rate varies significantly in the population. The rate of TL shortening ranges between 20 and 50 bp and is affected by factors such as the end-replication phenomenon, oxidative stress, and other DNA-damaging agents. In this review, we delve not only into the pathophysiology of TL shortening but also into its association with cardiovascular disease and the progression of atherosclerosis. We also provide current and future treatment options based on TL and telomerase function, trying to highlight the importance of these cutting-edge developments and their clinical relevance.

## 1. Introduction

During the last two decades, scientific society has been trying to investigate the role of potential biomarkers in the early detection of atherosclerotic disease [1]. The emphasis on inflammatory cytokines and their tight link with the progression of atherosclerosis, particularly in the coronaries, resulted in innovative diagnostic and therapeutic approaches [1,2,3,4,5]. However, even with access to the most advanced technology and most recently available secondary prevention therapies, the burden of recurrent events following acute coronary syndromes remains unacceptable, ranging from 10% to 20% in the first 12 months [6,7]. As such, atherosclerotic disease constitutes a global health and socioeconomic challenge, while the development of modalities for early detection of its progression and manifestations such as coronary artery disease (CAD) is considered mandatory.

The link between genetics and atherosclerosis has recently been added to the arsenal for estimating its progression. More particularly, physicians have suggested that information from genetic material may be the most specific biomarker for an individual’s predisposition to atherosclerotic disease progression and prognostication as well. This could be an important research endeavor since it is quite clear that the first signs of atherosclerosis may appear as early as adolescence, while severe complications appear later [8]. Observations showed that the length of the end sections of chromosomes may widely differ and be linearly associated with age and atherosclerotic burden. These DNA sequences, which can be visible in a light microscope, are called telomeres [9,10]. The length of telomeres seems to be generally affected by various inflammatory, stressed conditions and environmental factors [11]. In this review, we will explore the role of telomere length (TL) as a potential biomarker in the progression of atherosclerosis as well as cardiovascular disease.

## 2. Telomeres–Telomerase Function

Telomeres are the protective end caps of chromosomes that are essential for the preservation of our genome. Telomeres are regions that are constituted by millions of repeated DNA base pairs at the very end of each chromosome, which do not express any of the known proteins [12]. In humans and several other species, the ending telomeric repeat is represented by the sequence 5′-(TTAGGG) n-3′. The distal part of the telomere does not end in a double strand of DNA but the 5′-chain is shorter and a single-stranded G-rich section remains [12]. Telomerase function is not related to the existence of free 3′-chains, and no clear genetic explanation has been given. De Lange et al. showed that the free 3′-chain is constituted mainly by triplexes and quadruplexes of guanines (G-3, G-4, etc.) [13,14]. They suggested that this sequence may create a thermodynamic context for the interaction of a free 3′-chain with the double helix of telomeric DNA. As such, the related telomere forms a telomeric loop whose length is linearly correlated with TL in the various measurement methods [15].

Telomerase is a reverse transcriptase that uses a built-in RNA template to complete the end sequences of chromosome DNA. It is the protein part of our well-known human telomerase reverse transcriptase (hTERT) and human telomerase RNA component (hTERC) whereas a short region of telomerase is used for telomeric DNA synthesis [16]. Telomerase works as an enzyme of compound enzymes and proteins, such as pontin, reptin and chaperones heat shock protein 90 (HSP90) and TRiC [17]. Telomerase maturation takes place in Cajal bodies, which are rich in protein Telomerase Cajal body protein 1 (TCAB1) [18,19]. However, telomerase activity may be regulated in several stages including those of transcription, splicing, phosphorylation, maturation, and modifications of both hTERT and hTERC enzyme components. Several factors are responsible for the appropriate function of telomerase and the maintenance of TL, including the localization of telomerase in the cell nucleus or cytoplasm, the state of telomeric chromatin, changes in the packing of chromosome ends, etc. [20,21]. Other intriguing functions of telomerase have recently been discovered, with scientists attempting to unravel the underlying molecular pathways. It seems that telomerase upregulates the expression of NF-kappaB-dependent and glycolytic genes [22,23,24]. Additionally, modification of the inner functions of stem cells as well as epithelial–mesenchymal cells has been described while a protective role on mitochondrial DNA has been identified under oxidative-stressed conditions [23,25]. Interestingly, telomerase action seems to regulate the RNA Component of Mitochondrial RNA Processing Endoribonuclease (RMRP), whose mutations are responsible for Cartilage Hair Hypoplasia syndrome, while in another study normalization of the cell phenotype in patients with the inherited lipidosis of Niemann–Pick disease has been also observed [26,27]. As such, it is clear that the correct function of telomerase is not only associated with TL preservation, but also with the phenotype of a wide variety of genetic and metabolic disorders.

## 3. Telomere Length

Every person is born with a specific TL that ranges between 5 to 15 kb, which is affected as the years go by [28]. The rate of TL shortening hovers at 20–50 bp while it is dependent on several factors such as the end-replication phenomenon, oxidative stress, and other DNA-damaging agents [29,30,31,32]. According to the above-mentioned loop formation hypothesis, telomeric regions can form loops having a minimum length of about several thousand nucleotides, which can be used by the cell to quickly detect DNA breaks in this area [33]. By the end-replication phenomenon, a small telomeric DNA fragment is lost in every cell division due to the inability of transcription of the free 3′-chain. So naturally, telomeres reach a critical length and to such an extent that no loop can be formed. Telomere shortening is thought to be the cause of the restricted number of divisions in most human cells. Hayflick was the first to describe this occurrence and this phenomenon was later called the “Hayflick Limit” [34]. A DNA damage signal is received by the cell at this time, and telomeres lose their protective role. This critical shortening of TL leads the cell into senescence, and causative cell death, which is regulated by inner biochemical and pro-inflammatory changes via the transition of the cell into a senescence-associated secretory phenotype (SASP) [35,36]. The DNA-damage signal becomes permanent, leading to activation of cyclin-dependent inhibitor pathways, including either the p53/p21Cip or p16Ink4a/Rb while transcription factors such as nuclear factor kappa-light-chain-enhancer of activated B cells (NF-κB), CCAAT/enhancer-binding protein (C/EBP), and tumor protein p53 are controlling the procession [31,32,36]. Although TL varies along the tissue types due to the altered proliferation rates, a correlation has been observed between TL in different tissues and peripheral blood leucocytes [28,29,30,37,38]. As such, the leucocyte telomere length (LTL) has been considered as a surrogate marker of TL across the body and it would be the reference point of our review. 

## 4. LTL and Atherosclerosis

Telomeres, called the “biological clock” of cells, are a recognized marker of cell senescence. They are widely affected by a variety of intrinsic and environmental factors via the upregulation of oxidative stress levels. It is a condition in which increased levels of reactive oxygen species (ROS), such as superoxide anions, hydrogen peroxide, and hydroxyl radicals, are present due to a biological imbalance [39]. Dysfunctional mitochondria and immune cells are responsible for the main expression of ROS . The high guanine content of telomeres makes telomeres an easy target for ROS, leading to guanine oxidation. These point mutations and single or double DNA breaks affect telomeres’ function and cell proliferation [40,41]. Consequently, oxidative stress leads to TL shortening which is closely associated with tissue age-related decline regardless of the telomerase function [42]. As we discussed above, when a critical TL is reached, apoptotic mechanisms and molecular paths such as p53, MAP kinase (mitogen-activated protein kinase), and transcription factor kappa B are activated, leading to cell senescence [43]. Cells with SASP secrete a variety of pro- and inflammatory cytokines well known for their atherosclerotic effect [44]. SASP cells have been identified in vasculature regions with atherosclerotic plaques as well as in cardiomyocytes in biopsies of patients with heart failure [45,46,47]. In both conditions, the SASP phenotype was followed by significant short TL [45]. As a result, factors that induce oxidative stress and telomere shortening can lead to a vicious cycle that promotes a state of chronic inflammation, which causes vascular endothelial dysfunction and contributes to the development of atherosclerotic plaques.

Atherosclerosis is a multifactorial condition whose progression is affected by a variety of cardiovascular risk factors [2]. Recently, short LTL has been positively associated with cardiovascular risk factors such as high BMI, waist circumference, high levels of blood C-reactive protein, low levels of HDL (high-density lipoprotein), high levels of cholesterol and triglycerides, as well as insulin resistance and blood pressure [48,49,50]. In another study, although no association was observed between classical cardiovascular risk factors and short LTL, when the results were adjusted to smokers a strong statistical significance was detected [51]. Benetos et al. previously verified the link between hypertension and shorter telomeres [52]. In the same regard, Morgan et al. discovered that the exposed telomere ends led to arterial cell senescence in individuals with hypertension [53]. Finally, Haycock et al. concluded that patients with diabetes mellitus presented shorter LTL than their healthy counterparts [54]. Short telomeres have been linked to increased arterial stiffness, preclinical atherosclerosis, and poor diabetes management. This might be owing to the harmful consequences of persistent hyperglycemia and the accumulation of advanced glycation end products (AGEs) [55].

Atherosclerosis is characterized by the formation of plaques in the vessel wall. These emerge as a result of complex pathophysiological pathways involving pro- and anti-inflammatory cytokines [1,56]. Given the elevated cardiovascular risk in people with short LTL, it would be rational to assume that there is a connection between short LTL and subclinical atherosclerosis. Since the investigation of potential indicators of atherosclerosis progression constitutes a huge challenge worldwide, several studies are trying to detect the exact role of LTL in this context. Until now, the landscape has been quite hazy with the results from a number of studies being inconsistent. In 2016, a large study including 1459 middle-aged adults found no statistically significant link between shortened LTL and subclinical atherosclerosis [57,58]. Thus, TL does not seem to have a prognostic value in individuals without clinical signs of disease. Indeed, Nzietchueng and Nguyen et al. showed a significantly shorter TL in aortic cells with atherosclerotic lesions as well as in vasculature regions with low elasticity [59,60]. A possible explanation is that local vascular alterations affect TL as a result of oxidative stress. In the Framingham study, internal carotid artery intima-media thickness was linked with short LTL among obese males (BMI > 30 kg/m^2^) but not in the whole cohort. No association was observed between short LTL and common carotid artery intima-media thickness or carotid artery stenosis [61]. Trying to connect LTL with potential modifiable behavioral factors, Bountziouka et al. analyzed 422.797 patients from the UK Biobank. Although lifestyle changes appear to be quantitatively related with LTL, the magnitude of these effects is insufficient to appreciably affect the connection between LTL and various diseases or life expectancy [62]. In another study, Schellnegger et al. highlighted the detrimental role of sedentary life, not only in cardiovascular risk induction but also in LTL attrition. More specifically, a regular basis aerobic physical activity at a moderate to high level tends to help LTL preservation, while it is still unclear as to the optimal type and duration of exercising [63]. LTL was also not a significant predictor of intima-media thickness or plaque formation in the Asklepios study [57]. However, Panayiotou et al. showed an inverse association between LTL and common carotid artery intima-media thickness [64]. In the same aspect, the Strong Heart Study examined 2819 Americans without known cardiovascular risk factors for a follow-up period of 5.5 years to assess the predictive role of LTL in the occurrence and progression of carotid atherosclerosis. The shortest LTL had a 49% and 61% greater incidence of plaque formation and plaque development, respectively, than the longest LTL [65]. Finally, a strong association of carotid atherosclerosis with short LTL has been established in hypertensive patients, highlighting the unfavorable impact of hypertension on TL [66].

## 5. LTL and Cardiovascular Disease

Since observational studies illustrated that attrition of LTL is related to mortality, physicians are trying to investigate its role in cardiovascular disease (CVD), which is the leading cause of death worldwide [67]. The Hutchinson–Guilford Progeria Syndrome constitutes a striking example of age-related LTL shortening, in which the majority of the patients die from a myocardial infarction or stroke in their teenage years [68]. Premature senescence of fibroblasts as well as rapid TL shortening was observed in the cell cultures of these patients [68]. As such, there was a rationale for further exploration of LTL’s role in CVD, especially since studies showed that shorter LTL is associated with higher mortality rates. Indeed, in the Bruneck and LURIC studies, patients with a lower relative LTL presented higher death incidences in a 10-year follow-up [67,69,70]. In the same vein, multiple studies reported an increase in all-cause mortality risk, ranging from 17% to 66%, when patients with the longest telomeres were compared to subjects with the shortest telomeres [67,69,70,71,72]. However, some studies could not confirm the hypothesis of the association between LTL and CVD or mortality. This might be partially explained by the fact that some of these studies focused on low-risk populations with a modest number of events [73,74,75].

Several studies also investigated the association of LTL shortening with the presentation and progression of CAD. In two studies conducted in 2010 and 2021, physicians found that LTL in patients with stable CAD was significantly shorter than in healthy individuals of the same age (1.13 ± 0.52 CU in patients with CAD vs. 1.52 ± 0.81 CU in healthy individuals) which was further extended to sex analysis. It was interesting that men presented with shorter LTL than women, which could partially be related to the effect of estrogen, but this is an observation that remains hazy [70,76]. Trying to investigate the genetic background of premature CAD onset, Tian et al. compared Chinese patients with healthy individuals. It was shown that patients with premature CAD presented shorter LTL and higher circulating levels of oxidative stress components [77]. In another study, men with arterial hypertension, CAD, and early vascular aging (defined as arterial hypertension or CAD debut at young age—before 45 years, increased vascular wall stiffness according to the cardio-ankle vascular index), the LTL was significantly shorter than in men with arterial hypertension and CAD but without early vascular aging [78]. Haycock et al. conducted a meta-analysis on 43,000 individuals, including over 8000 patients with cardiovascular disease. It was demonstrated that, regardless of other risk factors, people with a shorter LTL were more likely to develop CAD [54]. In another meta-analysis published in 2020, shorter LTL was shown to be strongly connected to CAD severity, with Asians presenting the shortest LTL after ethnicity adjustment [79]. However, regarding the association of function stages of stable angina I–III (according to the Canadian Cardiovascular Society classifications) and LTL, no significant relation was observed [67,78].

One of the major manifestations of CAD is myocardial infarction (MI), and the development of biomarkers for early prognosis is referred to as mandatory [80,81,82,83,84]. LTL has been investigated as a potential biomarker with controversial results. A significantly shorter LTL has been observed in individuals with MI compared to healthy individuals even after adjustment for sex, body mass, and age [80]. Similarly, a study from the United Kingdom in 2017 demonstrated that in patients suffering from MI, LTL may be a useful prognostic biomarker for cardiovascular outcomes after the event, regardless of age. More particularly, MI patients with short LTL (defined as less than 0.96 CU in the study) presented significantly higher rates of all-cause and cardiovascular mortality within the first year after the event [85,86]. On the other hand, Russo et al. found no significant association between LTL and MI occurrence in young Italians [87]. Neither could Chan et al. confirm the hypothesis, as no statistically significant association was observed between relative LTL and adverse MI outcomes (death, recurrent MI, unplanned percutaneous coronary intervention revascularization, stroke, significant bleeding) in elderly Chinese patients, one year after their percutaneous coronary intervention [88]. Finally, while there was initially a connection between LTL and MI incidence in Czech women, the significance was lost after adjusting for major cardiovascular risk factors [86]. LTL has also been explored as a potential biomarker for the prognostication of stroke. The vast majority of the trials, which included over 37,000 and 25,000 people, respectively, could not establish a link between stroke risk and LTL [89,90]. However, a study in 2019 related short LTL with not only stroke incidence, but also post-stroke recovery in the elderly population [91] Findings of a Mendelian Randomization study in 2022 showed that longer telomeres were associated with decreased risk of several CVDs, including CAD, MI, and stroke, driving the rationale for further investigation in the domain (Figure 1) [92]. 

As such, we can safely conclude there is a need for further investigation of LTL’s role in cardiovascular disease. Although the results are encouraging, the lack of uniform assessment methodologies, as well as disparities in critical parameters such as patient age, ethnicity and race, and region of residence, prevent a safe conclusion from being reached.

## 6. A Target for Treatment

In Stockholm, during December 2009, the Nobel Prize in Physiology or Medicine was awarded to three biologists: Elizabeth H. Blackburn; Carol W. Greider; and Jack W. Szostak. Their project on how chromosomes are protected by telomeres and the enzyme telomerase changed the way that medicine used the therapeutic arsenal in several conditions including cardiovascular diseases [93,94,95]. They assumed that increasing telomerase activity can lead to telomere lengthening by influencing telomeres, telomerase function, and senescence. The hypothesis was that modest and potentially intermittent telomerase activity would allow cells to only repair telomeres, lowering the DNA damage response and SASP, while decreasing inflammation and oxidative stress levels [96].

Gene therapy using adeno-associated viruses to introduce the telomerase gene into aging mice appears to be promising, with incremental improvements in several biomarkers [96,97,98,99]. More particularly, beneficial effects were observed on insulin sensitivity, osteoporosis, neuromuscular coordination, and several molecular biomarkers of aging with a significant increase in median lifespan [98]. Another study by the same team supported that diet treatment with TA-65 (a product derived from a traditional Chinese medicinal plant—a weak telomerase activator) resulted in telomerase-dependent elongation of short telomeres and rescue of associated DNA damage, demonstrating that the TA-65 mechanism of action is via telomerase pathways [97]. Several studies are currently investigating the effect of TA-65 administration on cardiovascular health. Fernandez et al. showed that although the LTL was not significantly altered in patients receiving TA-65, there was an improvement in risk factors for cardiovascular disease. Reduced inflammatory levels were observed (low tumor necrosis factor-α levels) with a parallel reduction in body mass index, waist circumference, and atherosclerotic ratio LDL-C/HDL-C [99,100]. Finally, a promising ongoing trial (Phase II) is investigating whether a telomerase activator, TA-65, can reduce the proportion of senescent T cells in patients with acute coronary syndrome and confirmed CAD. It is also assessing the effect of TA-65 on decreasing telomere shortening, reducing oxidative stress, and improving endothelial function [101]. It is also possible to use therapeutic mRNAs that encode telomerase in senescent cells in specific regions of the vasculature [102]. The issue of adverse events due to the systematic administration of mRNAs can be solved by the use of modified nucleotides (on mRNA sequencing) that reduce innate immune response or functionalize nano- and microparticles to release therapeutic molecules directly to the inflamed endothelium [103]. Studies on cultured human cells revealed an increase in TL and a parallel ability for replication, while nanoparticles targeting endothelium are anticipated to lead in a slower or even reversible progression of vascular disease [102,103,104].

Senescent cells are subjected to immunosurveillance by multiple components of innate and adaptive immunity, including NK cells, T cells, and macrophages. Due to a decrease in immunosurveillance, senescent cells accumulate in aging and diseased tissues [105,106,107]. As a result, restoring or enhancing the immune system’s ability to precisely remove senescent cells may result in their effective clearance from tissues [105,106,107]. In experimental models, the removal of senescent cells led to a reduction in risk for atherosclerotic disease [8,108]. More particularly, the administration of senolytic drugs, Dasatinib and Quercetin (D & Q), led to improved left ventricular ejection fraction and fractional shortening. In the same way, the double administration of D & Q senolytic therapy reduced senescence burden and plaque calcification in transgenic mice [109,110]. Childs et al. showed a significant reduction in atherosclerotic formation and burden, as well as the mean lesion length, after administration of another senolytic drug, Navitoclax. In contrast to the reduction in CD8+ effector memory cells, this drug appeared to alleviate systemic inflammation or rejuvenation of progenitor pools, enhancing the presentation of naive CD8+ T cells [8,108,111].

Another side of the same coin is the use of senolytic drugs not only for removing senescent cells but for suppressing the processes leading to the development of SASP. The introduction of sequence-specific telomeric antisense oligonucleotides (tASOs) seems to alter the behavior of senescent cells when critical TL is reached [112]. These oligonucleotides, as the name implies, are similar to the 3′ overhangs of mammalian chromosomes and have shown strong anticancer effects in numerous cancer types, both in vivo and in vitro [112]. Finally, it would be very intriguing to see how hyperbaric oxygenation affects the progression of atherosclerosis, based on a hyperoxic–hypoxic paradox [113]. Studies showed that repeated intermittent hyperbaric oxygenation leads to significant lengthening of telomeres and a decrease in the number of senescent cells [112,113,114].

## 7. Considerations and Conclusions

Considering the above-mentioned innovative therapeutic methods, there are some concerns. Telomerase activation is the most prevalent hallmark of cancer cells (approximately 90%), and it is one of the most critical cancer indicators [115,116,117]. Telomerase activation in normal cells does not lead to carcinogenesis, while several cell types normally present high telomerase function (fetus, stem or progenitor, germ line in testes) [118]. For the formation of pre-cancerous cells, an accumulation of changes has to be synthesized with high telomerase activation. As such, it can be assumed that the induction of high telomerase activity in the aging cells of the organism, which may have gathered a substantial number of mutations, will result in malignant development. Therefore, the challenge for the physicians is to deploy a regulated telomerase activity allowing telomeres to reform, decreasing DNA damage response and SASP.

Another practical concern for using LTL directly in clinical practice is that current measurement accuracy is insufficient. Although measurement bias cannot be quantified in a population, this cannot be stated for individuals. The most prevalent approach, quantitative Polymerase Chain Reaction (qPCR), allows for quick testing but does not yield an absolute kilobase length estimate until combined with standard oligos [119]. Terminal Restriction Fragment (RTF) is another measurement method that provides mean length measure without recognition of individual short telomeres or missing ends. Finally, quantitative fluorescent in situ hybridization (Q-FISH) and primed in situ labeling (PRINS) are specialized for measuring telomeres in single cells and reporting the results as relative fluorescence units that exclusively relate to telomere length [120].

We are currently disregarding the potential effects of human telomere lengthening. To the best of our knowledge, shortening LTL seems to be a symptom of CAD rather than a causative factor. As such, the next step may be the use of LTL for the estimation of the biological age of the body, as well as the evaluation of the risk for future major cardiovascular adverse events. It will also be a promising alternative in examining the effectiveness of preventive medications. Further research is needed in the domain, especially in the role of telomerase function and its regulated activation. This is important not only in vascular beds but also in the therapeutic alternatives of different types of cancer. Physicians have to identify the lifestyle modifications connected with LTL preservation that contribute to increased life expectancy. One of these is the optimal intensity and duration of physical activity, as well as the type of exercise. This is a subject for extensive research due to the scourge of sedentary behavior and the Western lifestyle. Finally, research on the association of air pollution levels with LTL and major cardiovascular events, assessing patients from urban and rural areas, is needed to elucidate the impact of extrinsic factors on our cardiovascular health.

## Figures and Tables

**Figure 1 ijms-23-16010-f001:**
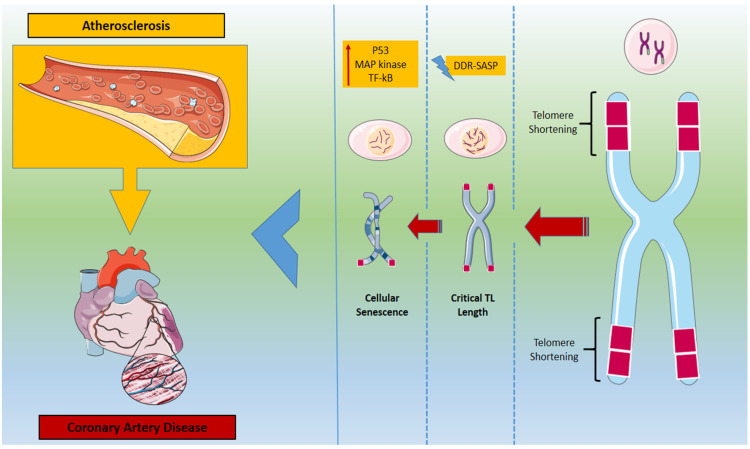
Telomere length shortening and its effect on atherosclerosis progression and coronary artery disease. DDR: DNA-damage-response; SASP: senescence-associated secretory phenotype; TF-kB: nuclear factor kappa-light-chain-enhancer of activated B cells.

## Data Availability

Not applicable.

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
