# Peer review of "Telomere Length: A Cardiovascular Biomarker and a Novel Therapeutic Target"

_ijms, 2022, doi:10.3390/ijms232416010_

Round 1
Reviewer 1 Report
In the submitted manuscript, the authors present an overview of the available information on research on the role of telomere length as a potential biomarker in the progression of atherosclerosis and cardiovascular diseases. Based on the available data, they state that leukocyte telomere length has been recognized as a surrogate marker of TL in the body. This fact became the basis for further considerations.
Telomerase activation is the most common characteristic of cancer cells and is one of the most critical indicators of cancer. Taking into account the presented facts, authors also show some concerns. In normal cells, activation of telomerase leads to carcinogenesis. However, in aging cells of organisms, it can lead to the development of a malignant tumor. Another problem with the direct use of leucocytes telomere length in clinical practice is insufficient measurement accuracy. Currently used tests (qPCR, RTF, etc.) do not provide sufficient data. Also, insufficient attention is currently paid to the potential effects of telomere lengthening in humans, with attention to possible adverse events. The authors clearly indicate the need for further research in this area.
To sum up, in my opinion it is a very valuable work that gives the possibility of a comprehensive look at the presented topic. At the same time, there is no specific indication for further research. And what is the 'novelty' of this manuscript really? This should be clearly marked.
I also have a note regarding Fig.1. - is it the authors' own drawing or is it a picture quoted from another publication? If it is a citation, it is necessary to enter a literature reference.
Reviewer 2 Report
I reviewed with interest the manuscript of Sagris et al. "Telomere Length: a Cardiovascular Biomarker and a Novel Therapeutic Target". In this article, the authors have provided an overview of the problem, highlighting the following sections: telomeres – telomerase function, telomere length, leucocytes telomere length and atherosclerosis, cardiovascular disease. The authors also discussed the possible target for treatment and some concerns about telomerase activation.
The authors managed to provide brief information on this problem, which can help the reader of the article to be acquainted with the state of the issue. I acknowledge that this topic is being actively studied and it is not possible to cover the entire amount of information in a particular article. Nevertheless, I suggest that the authors include a discussion of several more recent publications in the manuscript (1-5).
For example, a systematic review (1) showed that physical activity with regular moderate-to-high-intensity aerobic exercise appears to help maintain TL. It also discusses the optimal intensity, duration of physical activity, and type of exercise. Since reducing sedentary lifestyles can have a positive impact on maintaining and increasing TL, this review deserves inclusion in your manuscript.
A population-based study including 422,797 participants from the UK Biobank showed that although some potentially modifiable traits and healthy behaviors are quantitatively associated with LTL, these effects are not large enough to significantly change the association between LTL and various diseases or life expectancy. The authors of this article believe that attempts to change telomere length through lifestyle or behavioral changes may not provide significant clinical benefit (2).
In the article by Deng et al. the association of Telomere length with the risk of cardiovascular diseases is being studied using the current Mendelian randomization methods (3).
References:
1. Schellnegger M, Lin AC, Hammer N, Kamolz LP. Physical Activity on Telomere Length as a Biomarker for Aging: A Systematic Review. Sports Med Open. 2022 Sep 4;8(1):111. doi: 10.1186/s40798-022-00503-1.
2. Bountziouka V, Musicha C, Allara E, Kaptoge S, Wang Q, Angelantonio ED, Butterworth AS, Thompson JR, Danesh JN, Wood AM, Nelson CP, Codd V, Samani NJ. Modifiable traits, healthy behaviours, and leukocyte telomere length: a population-based study in UK Biobank. Lancet Healthy Longev. 2022 May;3(5):e321-e331. doi: 10.1016/S2666-7568(22)00072-1.
3. Deng Y, Li Q, Zhou F, Li G, Liu J, Lv J, Li L, Chang D. Telomere length and the risk of cardiovascular diseases: A Mendelian randomization study. Front Cardiovasc Med. 2022 Oct 24;9:1012615. doi: 10.3389/fcvm.2022.1012615
4. Daios S, Anogeianaki A, Kaiafa G, Kontana A, Veneti S, Gogou C, Karlafti E, Pilalas D, Kanellos I, Savopoulos C. Telomere Length as a Marker of Biological Aging: A Critical Review of Recent Literature. Curr Med Chem. 2022;29(34):5478-5495. doi: 10.2174/0929867329666220713123750.
5. Armstrong ND, Irvin MR, Haley WE, Blinka MD, Kamin Mukaz D, Patki A, Rutherford Siegel S, Shalev I, Durda P, Mathias RA, Walston JD, Roth DL. Telomere shortening and the transition to family caregiving in the Reasons for Geographic and Racial Differences in Stroke (REGARDS) study. PLoS One. 2022 Jun 3;17(6):e0268689. doi: 10.1371/journal.pone.0268689.
